# Enhancing Biological Reasoning in LLMs via Synthetic Reasoning Traces for Cellular Perturbation Prediction

## Abstract

Predicting cellular responses to genetic perturbations represents a fundamental challenge in systems biology, critical for advancing therapeutic discovery and virtual cell modeling. While large language models (LLMs) show promise for biological reasoning, their application to perturbation prediction remains underexplored due to challenges in adapting them to structured experimental data. We present SynthPert, a novel method that enhances LLM performance through supervised fine-tuning on synthetic reasoning traces generated by frontier models. Using the PerturbQA benchmark, we demonstrate that our approach not only achieves state-of-the-art performance but surpasses the capabilities of the frontier model that generated the training data. Our results reveal three key insights: (1) Synthetic reasoning traces effectively distill biological knowledge even when partially inaccurate, (2) This approach enables cross-cell-type generalization with 87% accuracy on unseen RPE1 cells, and (3) Performance gains persist despite using only 2% of quality-filtered training data. This work shows the effectiveness of synthetic reasoning distillation for enhancing domain-specific reasoning in LLMs.

## 1 Introduction

Accurately predicting cellular responses to genetic perturbations represents a fundamental challenge in systems biology, with critical applications from drug discovery to virtual cell modeling (Bunne et al., 2024). While deep learning approaches like GEARS (Roohani et al., 2024a) and scGPT (Cui et al., 2024) have advanced perturbation prediction, they struggle with generalization to unseen biological contexts—a key requirement for real-world applicability. Recent breakthroughs in large language models (LLMs) offer new paradigm through their ability to reason over biological knowledge, yet their potential remains underexploited for this task.

The rapid evolution of LLM reasoning capabilities, driven by architectural innovations (Guo et al., 2025) and techniques like chain-of-thought prompting (Wei et al., 2022), has enabled novel scientific applications. Projects like AI Co-Scientist (Gottweis et al., 2024) demonstrate LLMs' ability to assist hypothesis generation, while benchmarks like LAB-Bench (Laurent et al., 2024) assess practical biological reasoning. However, existing evaluations focus on literature analysis and experimental design, neglecting the core challenge of predicting intervention outcomes in complex cellular systems.

PerturbQA (Wu et al., 2024) addresses this gap by reformulating perturbation experiments into natural language tuples (cell type, perturbation, gene) → {up, down, unperturbed}. While this three-class formulation simplifies the complexity of gene expression dynamics, it provides a practical framework grounded in statistical hypothesis testing for actionable biological predictions. However, current approaches simplify the task through problematic decomposition: first predicting differential expression, then directionality only for perturbed genes. This contrasts with real-world scenarios where researchers lack prior knowledge of perturbation effects, requiring joint three-class prediction.

Our work makes three key contributions:

- **Synthetic Reasoning Distillation**: A novel method enhancing LLMs through fine-tuning on generated chain-of-thought explanations rather than raw experimental data

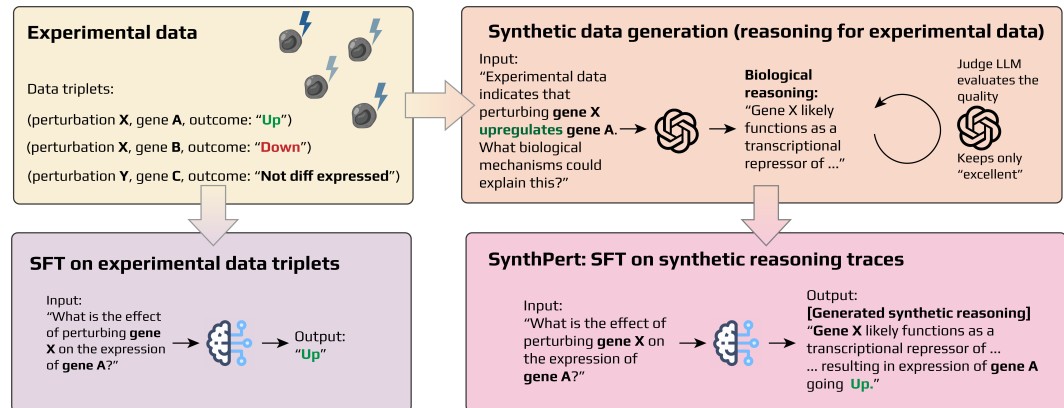

Figure 1: Illustration of the SynthPert workflow. Given experimental perturbation data in the form of "(*perturbation, gene, outcome*)" data tuples (top left panel), our goal is to create an LLM capable of predicting responses to unseen perturbations. We consider two strategies: (i) SFT on experimental data directly (bottom left panel), and (ii) a synthetic chain-of-thought based supervised fine-tuning. The arrows between panels indicate information flow. In particular, the latter involves experimental data *indirectly*, in the process of creating synthetic reasoning traces for given data tuples, using a frontier LLM. A separate judge LLM evaluates their quality, and keeps only those synthetic explanations that were graded "excellent". Finally, we fine-tune the base LLM on the generated chain-of-thought explanations.

- **Generalizable Prediction**: State-of-the-art performance (78% AUROC) with successful cross-cell transfer (87% accuracy on unseen RPE1 lineages)
- **Practical Task Formulation**: Direct three-class prediction without artificial task decomposition, better matching biological use cases

Central to our approach is the insight that reasoning structure, rather than factual accuracy, drives biological generalization. By filtering synthetic explanations from frontier models (o4-mini) through quality critics, we enable smaller LLMs to surpass teacher models' performance—a distillation paradox revealing untapped potential in pretrained biological schemas. Our results challenge conventional wisdom through three findings: (1) Structured explanations improve minority class prediction despite data imbalance, (2) External biological databases degrade performance when forced into reasoning paths, and (3) Practical three-class prediction outperforms decomposed binary tasks, better matching real-world use cases.

We evaluate our approach on the PerturbQA benchmark, demonstrating state-of-the-art performance and strong generalization to unseen cell types. Our analysis reveals insights about the role of reasoning structure in biological prediction tasks and establishes synthetic reasoning distillation as an effective approach for domain-specific LLM enhancement.

## 2 BACKGROUND

### 2.1 CLASSICAL DEEP LEARNING APPROACHES FOR CELLULAR PERTURBATION PREDICTION

Predicting cellular perturbation outcomes has been an active area of method development to infer interventional distributions over gene expression vectors (Lotfollahi et al., 2023; Lopez et al., 2023; Gebregziabher et al., 2024). Of particular interest, is to predict *unseen* pert outcomes, and this requires introduction of some notion of prior knowledge.

**GEARS** Roohani et al. (2024b) proposed GEARS, a graph neural network that incorporates structured biological prior knowledge through gene co-expression graphs and gene ontology networks. This enables generalization to perturbations involving genes absent from training data by exploiting connectivity patterns between seen and unseen genes.

**Single-cell foundation models**   Single-cell transformer models such as scGPT (Cui et al., 2024) take a different approach by pre-training on large-scale gene expression atlases and tokenizing gene expression vectors. Despite their sophisticated architecture and a large number of parameters, these models often struggle to outperform much simpler baselines for perturbation prediction tasks (Ahlmann-Eltze & Theis, 2024; Kernfeld et al., 2024), suggesting limitations in their ability to model perturbation data effectively.

## 2.2   LLM-based approaches for Cellular Perturbation Biology

More recently, large language models have emerged as a promising alternative for modeling perturbation biology, offering unique capabilities to incorporate biological knowledge and reasoning.

**LLM-derived gene embeddings**   GenePT (Chen & Zou, 2024) proposed an approach that leverages LLMs to embed gene or protein summaries from scientific literature and using these embeddings in downstream modelling, such as for creating cell embeddings from gene expression data. By generating gene embeddings from NCBI text descriptions, GenePT captures rich semantic information about gene functions and relationships. These embeddings have shown relatively strong performance when combined with ML models such as Gaussian Processes for perturbation prediction (Märtens et al., 2024), demonstrating that LLMs can indeed effectively encode biologically relevant prior knowledge that generalizes to unseen perturbations.

**LLM-based reasoning frameworks**   SUMMER (Wu et al., 2024) introduced a retrieval-augmented generation (RAG) framework for perturbation biology, employing LLMs to: (1) summarize gene descriptions, (2) retrieve similar seen perturbations via knowledge graph proximity, and (3) reason through structured prompting. This approach outperformed previous methods on the PerturbQA benchmark without fine-tuning, thus providing a natural framework we build upon.

However, SUMMER notably simplified the problem by converting the natural three-way classification task into two separate binary classifications. First, they classified genes as differentially expressed or not, then predicted directionality ("up" or "down") only for the subset of genes already identified as differentially expressed. While this decomposition proved effective, in practical applications researchers typically don't have advance knowledge of which genes will be differentially expressed, making it easier to work with a 3-class prediction model.

**Limitations of current LLM approaches**   While showing promise, current LLM-based methods primarily operate at inference time rather than systematically learning biological reasoning patterns. GenePT uses LLMs solely as embedding generators, while SUMMER relies on a retrieval-based approach that may struggle with perturbation scenarios that lack similar examples in its retrieval database. Neither approach fully leverages the reasoning capabilities of LLMs through targeted fine-tuning on perturbation biology tasks. This suggests an opportunity to develop models that can learn from perturbation data directly while maintaining the interpretability and knowledge integration advantages of LLMs—the direction we pursue in this work.

## 3   Method: SynthPert

In this work, we develop an LLM-based perturbation prediction model that, given a *cell type*, *perturbation*, and *gene of interest*, predicts whether the gene is up-regulated, down-regulated, or not differentially expressed. Formally, we learn the mapping:

$$(\text{cell type}, \text{perturbation}, \text{gene}) \mapsto \{\text{``up''}, \text{``down''}, \text{``not differentially expressed''}\}$$

parameterized by an LLM. Although predictions are made for individual genes, our approach enables genome-wide in silico screening through systematic application across all genes.

Unlike Wu et al. (2024), who predicted directionality (i.e., "up" or "down") only for genes pre-identified as differentially expressed, our approach directly classifies all three possible states simultaneously. This comprehensive formulation better matches real-world biological workflows where researchers lack prior knowledge of perturbation effects.

## 3.1 SYNTHETIC TRAINING STRATEGIES

While existing work uses pretrained LLMs without modification, we explore two complementary fine-tuning approaches:

**SFT on Data Tuples (Baseline)**   The straightforward approach involves direct supervised fine-tuning (SFT) on observed (input, output) data tuples. While we consider this primarily as a baseline, to our knowledge this represents the first application of SFT to perturbation prediction. This method trains the model to associate inputs with outcomes without explanatory context.

**SFT on Synthetic Explanations (SynthPert)**   Our novel strategy focuses on enhancing biological reasoning through synthetic data generation. Rather than training directly on experimental tuples, we use a frontier model to generate mechanistic explanations for observed outcomes, then fine-tune on these reasoning traces. This indirect approach teaches causal relationships rather than surface-level associations.

## 3.2 SYNTHETIC CHAIN-OF-THOUGHT GENERATION

As illustrated in Figure 1, our workflow generates biological explanations through two complementary strategies:

**Approach 1 (Prediction + Explanation):** The frontier model (OpenAI o4-mini) receives only the cell type, perturbation, and gene name, requiring it to both predict the outcome and generate supporting reasoning. We retain only traces where the model correctly predicted the outcome.

**Approach 2 (Explanation from Outcome):** Here we provide the ground truth outcome alongside the input tuple, tasking the model with generating a mechanistic rationale. A separate critic model (also a frontier model) grades explanations on a 5-point scale: 'excellent', 'good', 'average', 'bad', or 'terrible'. We retain only 'excellent' graded explanations.

Both approaches allow the model to respond with "I don't know" to reduce hallucinations. Approach 2 produced higher quality explanations (see Supplementary Material) and forms our final implementation.

## 3.3 SUPERVISED FINETUNING IMPLEMENTATION

We perform SFT using Low-Rank Adaptation (LoRA) (Hu et al., 2022) on a DeepSeek-R1 8B model (Guo et al., 2025). Despite generating traces for only a subset of training data initially – filtered to a small fraction via quality control – SYNTHPERT achieves superior performance to models trained on full datasets, demonstrating remarkable data efficiency.

For rigorous comparison, we implement a LLM + SFT ON DATA baseline using identical hyperparameters but training directly on perturbation tuples without explanations. This controlled experiment isolates the value added by synthetic reasoning traces versus mere exposure to experimental data.

## 4 RESULTS

### 4.1 EXPERIMENTAL SETUP

**Datasets**   For evaluation, we use the PerturbQA benchmark released by Wu et al. (2024). This is a relatively comprehensive data source, spanning Perturb-seq experiments across four cell lines: K562, RPE1, HepG2, and Jurkat, derived from data by Replogle et al. (2022) and Nadig et al. (2024).

In PerturbQA, for every cell line, we have a number of tuples "(*perturbation, gene, outcome*)". While the outcome label is one of {"up", "down", "not differentially expressed"}, the benchmark separates two tasks: 1) differential expression, and 2) direction of change, where the latter task has been subsetted to those data tuples that are indeed differentially expressed. In total, the dataset contains $N = 84,550$ examples for the direction of change task and $N = 614,479$ for the differential expression task. These have been split into a 75% train and 25% test sets. A detailed breakdown by cell line is available in Table 5 in the Appendix.

**Implementation details** For all SYNTHPERT results reported in this paper, we supervised fine-tuned on synthetic data created via the critic-based method (our second approach described in Section 3.1, and as illustrated in Figure 1), as we saw similar results across both methods but this approach is more data efficient as the critic filters out low-quality explanations before fine-tuning.

**Baselines** Following the PerturbQA original data splits, we compare our "LLM + SFT ON DATA" baseline and SYNTHPERT with the baselines considered by Wu et al. (2024). Specifically, we include PHYSICAL, which naively predicts differential expression based on known physical interactions between genes from STRINGDB (Szklarczyk et al., 2021), and GAT (Veličković et al., 2018), a graph attention network trained on biological knowledge graphs with a ternary classification objective. We also compare against state-of-the-art methods previously described in the background section: GEARS (Roohani et al., 2024a), scGPT (Cui et al., 2024), and GenePT (Chen & Zou, 2024). Additionally, we evaluate against SUMMER (Wu et al., 2024), a RAG-based approach that leverages knowledge graphs and LLM reasoning to predict perturbation effects in biological systems.

**Evaluation Protocol** We evaluate our models using two distinct protocols. First, we follow the original PerturbQA setup as proposed by Wu et al. (2024), adhering to their data splits and AUROC metric for the two separate tasks (differential expression and direction of change), as detailed in Section 4.2.

Second, we introduce a more unified evaluation approach that directly addresses the three-class classification problem {"up", "down", "not differentially expressed"}, which better aligns with our model's design. For this three-class evaluation, we report precision, recall, and F1 scores for each class, along with overall accuracy, providing a more comprehensive assessment of model performance across the imbalanced class distribution. This approach allows for a more comprehensive assessment of SYNTHPERT's performance relative to both the base LLM and existing methods.

Furthermore, alongside the original PerturbQA split, we propose an additional evaluation protocol where an entire cell type (RPE1) is held out as a test set. This cross-cell-type evaluation provides a more rigorous assessment of generalization capabilities across biological contexts, testing whether models can transfer knowledge to previously unseen cellular environments, as detailed in Section 4.3.

**Baseline Rationale** The motivation for including the "LLM + SFT ON DATA" baseline is two-fold. First, we aim to quantify the utility of our synthetic chain-of-thought data relative to the information content directly present in the observed experimental data. Second, when treated as a three-class problem, the data distribution is highly imbalanced—the "not differentially expressed" category significantly dominates the other two classes. Therefore, we needed to ensure that SYNTHPERT is learning meaningful biological relationships and not simply memorizing the underlying data distribution. The "LLM + SFT ON DATA" baseline, trained directly on the experimental data, provides a control that would demonstrate performance achievable through distribution learning alone.

In Section 4.3, we also include the frontier model OpenAI o4-mini as an additional baseline. This is the same model we used for generating synthetic chain-of-thought traces, providing a direct comparison between the teacher model and our fine-tuned LLM. This comparison is particularly meaningful as it helps evaluate whether our approach can effectively distill and enhance the biological reasoning capabilities of the frontier model. Generally, one would not expect a much smaller (8 billion parameter) model to match or exceed the capabilities of a large frontier model on complex reasoning tasks. However, this comparison allows us to test the hypothesis that targeted fine-tuning on high-quality synthetic reasoning chains might enable more efficient utilization of parameter capacity for specific biological reasoning tasks. The comparison also examines whether frontier models contain implicit knowledge about biological mechanisms that can be extracted and refined through our synthetic data approach, potentially allowing a smaller model to achieve strong performance on this specialized task.

**Model Configuration** In all experiments, we use a DeepSeek-R1 distilled Llama 3.1 8B model (Team et al., 2024; Guo et al., 2025). Importantly, we fine-tune the model only once and evaluate it on both tasks (differential expression and directionality prediction), demonstrating the versatility of our approach.

## 4.2 DIFFERENTIAL EXPRESSION AND DIRECTION OF CHANGE

Table 1 presents the comparison of AUROC values on the PerturbQA Differential Expression task across all four cell lines. While SUMMER achieves the highest performance among existing methods, SYNTHPERT substantially and significantly outperforms it across all cell lines. Notably, SYNTHPERT demonstrates remarkable improvements on the RPE1 cell line (AUROC increase from 0.58 to 0.78) and Jurkat cell line (increase from 0.58 to 0.79).

Table 1: Results on Differential Expression as binary prediction, following the original PerturbQA evaluation (Wu et al., 2024). AUROC is computed over the predictions associated with each gene, and averaged over perturbations.

| Task | Model | K562 | RPE1 | HepG2 | Jurkat |
|------|-------|------|------|-------|--------|
| | PHYSICAL | 0.53 | 0.52 | 0.52 | 0.54 |
| | GAT | 0.55 | 0.57 | 0.57 | 0.55 |
| | GEARS | 0.54 | 0.50 | 0.48 | 0.51 |
| | SCGPT | 0.52 | 0.52 | 0.48 | 0.51 |
| Differential | GENEPT-GENE | 0.57 | 0.54 | 0.55 | 0.55 |
| expression | GENEPT-PROT | 0.57 | 0.56 | 0.54 | 0.55 |
| | SUMMER | 0.60 | 0.58 | 0.61 | 0.58 |
| | LLM + SFT ON DATA | 0.59 | 0.59 | 0.60 | 0.60 |
| | SYNTHPERT | **0.70** | **0.78** | **0.74** | **0.79** |

Importantly, our "LLM + SFT ON DATA" baseline performs significantly worse than SYNTHPERT, confirming that naive fine-tuning with the training data alone would not have achieved these performance gains. This comparison clearly demonstrates the added value of our synthetic chain-of-thought generation workflow in enhancing the model's biological reasoning capabilities.

Table 2 presents results for the Direction of Change prediction task. SYNTHPERT demonstrates strong performance on this task as well, despite being fine-tuned on three-class reasoning traces. For this evaluation, we simply adjust the prompt to inform the model that the gene is differentially expressed and ask it to predict one of the two directional labels. SYNTHPERT outperforms SUMMER on three out of four cell lines, while performing comparably on the fourth (Jurkat). Consistent with our previous findings, the "LLM + SFT ON DATA" baseline performs substantially worse across all four cell lines, further confirming that extracting meaningful patterns from perturbation data requires more sophisticated approaches than direct fine-tuning.

Table 2: Results on Direction of Change as binary prediction, following the original PerturbQA evaluation (Wu et al., 2024). AUROC is computed over the predictions associated with each gene, and averaged over perturbations.

| Task | Model | K562 | RPE1 | HepG2 | Jurkat |
|------|-------|------|------|-------|--------|
| | GAT | 0.58 | 0.60 | 0.64 | 0.59 |
| | GEARS | 0.64 | 0.60 | 0.52 | 0.51 |
| | SCGPT | 0.48 | 0.53 | 0.51 | 0.51 |
| Direction | GENEPT-GENE | 0.53 | 0.57 | 0.58 | 0.57 |
| of change | GENEPT-PROT | 0.57 | 0.57 | 0.55 | 0.58 |
| | SUMMER | 0.62 | 0.64 | 0.65 | **0.66** |
| | LLM + SFT ON DATA | 0.47 | 0.55 | 0.55 | 0.50 |
| | SYNTHPERT | **0.65** | **0.73** | **0.72** | 0.65 |

## 4.3 DIRECT THREE-CLASS PERTURBATION EFFECT PREDICTION

Having established the strong performance of SYNTHPERT on the individual binary tasks in the PerturbQA benchmark, we now evaluate its capabilities on the more challenging and biologically

relevant direct three-class prediction problem. Tables 3 and 4 present detailed performance metrics across all three possible gene expression states.

Table 3 shows that SYNTHPERT significantly outperforms all baselines in terms of overall accuracy (89% versus 15% for the base LLM and 52% for both the SFT baseline and o4-mini). Notably, SYNTHPERT substantially outperforms o4-mini—the very model used to generate its training data—demonstrating that our synthetic fine-tuning approach effectively distills and refines the biological reasoning capabilities of the larger model. While o4-mini achieves the highest recall for upregulated genes (0.62 compared to SYNTHPERT's 0.14), it suffers from poor precision (0.12), suggesting it tends to over-predict this class. In contrast, SYNTHPERT shows more balanced performance with significantly higher precision across all classes.

Interestingly, SYNTHPERT shows a different precision-recall trade-off compared to the base LLM for upregulated genes. SYNTHPERT achieves the highest precision (0.49 compared to only 0.07 for the base LLM) but lower recall (0.14) compared to both the base LLM (0.32) and o4-mini (0.62). This suggests that SYNTHPERT has learned to be more selective in its predictions, only identifying the most confident cases of upregulation. In contrast, the other models appear to overpredict upregulation, capturing more true positives but at the cost of numerous false positives—a pattern consistent with insufficient understanding of the biological mechanisms governing gene upregulation. Despite this trade-off, SYNTHPERT still achieves a better F1 score for upregulated genes than the base LLM (0.22 vs 0.11), indicating that its precision gains outweigh its recall losses in this balanced metric.

The dramatic improvement in overall accuracy is primarily driven by SYNTHPERT's exceptional performance in correctly identifying non-differentially expressed genes (F1 score of 0.95 compared to o4-mini's 0.68 and base LLM's 0.21), which constitute the majority class in most perturbation datasets. Beyond the majority class, SYNTHPERT also demonstrates substantially better performance on downregulated genes, with an F1 score of 0.53—nearly triple the performance of both o4-mini and the SFT baseline (0.18). This suggests our approach not only distills knowledge from the teacher model but actually enhances it through the focused learning on high-quality synthetic reasoning examples.

Table 3: Performance comparison on direct three-class perturbation effect prediction. All perturbations evaluated are unseen during training. Results show precision, recall, and F1 scores for each class (Not Differentially Expressed, Upregulated, and Downregulated), along with overall accuracy across all models.

| Model | Not Diff. Expressed | | | UP Regulated | | | DOWN Regulated | | | Accuracy |
|-------|-------|------|------|-------|------|------|-------|------|------|----------|
|       | Prec. | Rec. | F1   | Prec. | Rec. | F1   | Prec. | Rec. | F1   |          |
| Base LLM | 0.87 | 0.12 | 0.21 | 0.07 | 0.32 | 0.11 | 0.08 | 0.38 | 0.14 | 0.15 |
| SFT on data | 0.90 | 0.55 | 0.69 | 0.08 | 0.29 | 0.13 | 0.12 | 0.35 | 0.18 | 0.52 |
| o4-mini | **0.92** | 0.54 | 0.68 | 0.12 | **0.62** | 0.20 | 0.15 | 0.23 | 0.18 | 0.52 |
| SYNTHPERT | **0.92** | **0.97** | **0.95** | **0.49** | 0.14 | **0.22** | **0.52** | **0.53** | **0.53** | **0.89** |

Table 4: Cross-cell-type generalization performance with RPE1 cell line held out. Models are fine-tuned exclusively on data from HepG2, Jurkat, and K562 cell lines, then evaluated on the unseen RPE1 cell line to assess transfer learning capabilities across biological contexts

| Model | Not Diff. Expressed | | | UP Regulated | | | DOWN Regulated | | | Accuracy |
|-------|-------|------|------|-------|------|------|-------|------|------|----------|
|       | Prec. | Rec. | F1   | Prec. | Rec. | F1   | Prec. | Rec. | F1   |          |
| Base LLM | 0.87 | 0.12 | 0.22 | 0.05 | **0.29** | 0.09 | 0.09 | 0.34 | 0.14 | 0.15 |
| SFT on data | 0.87 | 0.43 | 0.58 | 0.06 | 0.24 | 0.09 | 0.10 | 0.19 | 0.14 | 0.40 |
| o4-mini | 0.91 | 0.54 | 0.68 | 0.09 | 0.20 | 0.13 | 0.13 | **0.53** | 0.21 | 0.52 |
| SYNTHPERT | **0.92** | **0.96** | **0.94** | **0.31** | 0.16 | **0.21** | **0.49** | 0.41 | **0.45** | **0.87** |

Table 4 presents results from our more challenging cross-cell-type generalization experiment, where models trained on HepG2, Jurkat, and K562 cell lines are evaluated on the completely unseen RPE1

cell line. SYNTHPERT maintains remarkably strong performance even in this zero-shot transfer setting, with only a modest drop in overall accuracy from 89% to 87%. This robust generalization suggests that SYNTHPERT has learned transferable biological principles rather than simply memorizing patterns specific to individual cell types.

Comparing SYNTHPERT to o4-mini in this cross-cell-type scenario reveals particularly interesting insights. Despite being fine-tuned on synthetic data generated by o4-mini, SYNTHPERT significantly outperforms its teacher model in overall accuracy (87% vs. 52%). This substantial gap demonstrates that our synthetic fine-tuning approach not only distills knowledge but enhances generalization capabilities across different biological contexts. While o4-mini achieves the highest recall for downregulated genes (0.53), surpassing even SYNTHPERT (0.41), its precision is considerably lower (0.13 vs. 0.49), resulting in a weaker F1 score (0.21 vs. 0.45). This pattern suggests that o4-mini tends to over-predict gene downregulation in novel cellular contexts, while SYNTHPERT makes more calibrated predictions.

Across all classes, SYNTHPERT demonstrates better balanced performance than the baselines, with particularly dramatic improvements in identifying non-differentially expressed genes (F1 score of 0.94 compared to o4-mini's 0.68 and the base LLM's 0.22). For minority classes, SYNTHPERT consistently favors precision over recall, making fewer but more reliable predictions—a valuable characteristic for practical experimental design.

Manual evaluation of reasoning traces by domain experts (detailed in Appendix B) revealed that chains-of-thought leading to correct predictions had higher factual accuracy (0.92 vs 0.76 for incorrect predictions), and relied more on gene-level functional descriptions rather than high-level pathway categorizations. This suggests that SYNTHPERT succeeds by learning mechanistic relationships between specific gene functions rather than exploiting superficial biological abstractions.

These results collectively demonstrate that SYNTHPERT's synthetic fine-tuning approach enables effective transfer learning across fundamentally different cell types with distinct biological characteristics. This generalization capability is particularly noteworthy given that RPE1 cells (retinal pigment epithelial cells) have significantly different biological functions compared to the training cell types (liver cancer cells, leukemia cells, and chronic myelogenous leukemia cells). The model's ability to maintain strong performance across these diverse cellular contexts suggests it has captured fundamental regulatory principles that transcend cell-type-specific patterns.

## 5 DISCUSSION

Our work demonstrates that synthetic reasoning traces enable LLMs to overcome their pretraining limitations in biological prediction tasks. SYNTHPERT establishes three key advances in this domain.

First, our synthetic reasoning approach proves more effective than direct training on experimental data. By fine-tuning on quality-filtered chain-of-thought (CoT) explanations rather than raw experimental data, we achieve state-of-the-art PerturbQA performance (78% AUROC) while using only 2% of the potential training data. This finding confirms that it's the structured reasoning patterns that drive accurate perturbation prediction. The efficiency gains suggest that targeted reasoning enhancement can be more effective than simply increasing dataset size.

Second, we demonstrate effective cross-cell-type generalization through synthetic explanations. SYNTHPERT's 87% accuracy on unseen RPE1 cells, compared to 40% for direct supervised fine-tuning on experimental data, reveals that synthetic traces encode transferable biological principles rather than cell-specific patterns. This suggests the model has learned fundamental mechanisms of gene regulation that apply across different cellular contexts.

Third, we observe a striking result where SYNTHPERT substantially surpasses its teacher model o4-mini (89% vs 52% accuracy) despite being trained on synthetic data generated by that same model. This is particularly noteworthy given that the base model initially achieved only 15% accuracy. The substantial improvement suggests that targeted fine-tuning on high-quality synthetic reasoning traces can unlock latent capabilities in smaller models for domain-specific tasks.

This finding supports the Superficial Alignment Hypothesis (Zhou et al., 2023): limited fine-tuning data (14k examples) suffices because pretraining on scientific text embeds biological reasoning

capabilities that targeted CoT traces activate. This suggests that teaching models to reason through biological mechanisms can be highly effective for domain-specific tasks.

## 5.1 LIMITATIONS AND CHALLENGES

Three technical limitations emerge from our analysis. First, class imbalance effects create challenges similar to those seen in long-tailed classification (ValizadehAslani et al., 2024). The precision-recall tradeoff for upregulated genes (0.49 vs 0.14) reflects PerturbQA's skewed label distribution, where 85% of genes are non-differentially expressed. However, our LLM + SFT ON DATA baseline, which could have benefited from exposure to the class distribution, performed substantially worse than SYNTHPERT (52% vs 89% accuracy). To further verify that our results weren't due to memorizing class proportions, we rebalanced the SYNTHPERT training data such that each class represented exactly a third of the training data. The resulting model performed roughly the same as the original, confirming that performance gains stem from the reasoning structure rather than class distribution learning.

Second, our attempts to incorporate external biological knowledge were counterproductive. Adding EnrichR pathway analysis (Kuleshov et al., 2016) during inference degraded performance ($\Delta$AUROC = -0.07), likely due to input length saturation (Liu et al., 2023). This suggests that explicit biological knowledge injection may interfere with LLMs' implicit reasoning capacities.

Third, validating the biological accuracy of CoT rationales remains challenging. While we provide manual verification of reasoning traces for a small sample (Appendix B), this process is extremely time-consuming as it requires evaluating each biological claim within the reasoning chain rather than just the final prediction. This represents an important consideration for LLMs in scientific discovery applications (Ji et al., 2023).

## 5.2 FUTURE DIRECTIONS

Our findings suggest several promising research directions for enhancing biological reasoning in language models. Implementing Reinforcement Learning from Biological Feedback could significantly improve model performance, potentially through approaches like GRPO (Shao et al., 2024) with rewards derived from pathway database consistency checks against resources like STRING (Szklarczyk et al., 2021). This would allow models to receive direct feedback on the biological plausibility of their reasoning chains, potentially reducing hallucinated mechanisms while reinforcing scientifically valid explanations.

Beyond RL, we see opportunities in Multi-Task Co-Distillation by jointly training models on perturbation prediction and biomedical question answering datasets such as MedQA (Jin et al., 2020), enabling knowledge transfer across related biological reasoning tasks. Additionally, the computational efficiency of our approach could be further improved by applying recent LoRA variants (Hayou et al., 2024) to reduce memory overhead during Chain-of-Thought synthesis, making these approaches more accessible to researchers with limited computational resources while potentially improving parameter efficiency.

## 5.3 CONCLUSION

SYNTHPERT establishes synthetic reasoning distillation as an efficient paradigm for biological LLMs, achieving state-of-the-art prediction and unprecedented cross-cell generalization. Our results challenge the prevailing assumption that biological AI requires massive experimental datasets – instead, carefully structured reasoning traces unlock pretrained knowledge through targeted activation. For ML practitioners, this work offers a blueprint for domain-specific reasoning enhancement; for biologists, a step toward interpretable in silico experimentation.

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

# A   APPENDIX

## A   DATASET STATISTICS

Table 5: Number of data points across tasks and cell lines in PerturbQA.

| Task | HepG2 | Jurkat | K562 | RPE1 | Total |
|------|-------|--------|------|------|-------|
| Direction of Change | 17,860 | 20,058 | 19,980 | 26,652 | 84,550 |
| Differential Expression | 126,889 | 142,822 | 157,679 | 187,089 | 614,479 |

## B   BIOLOGICAL FACTUALITY OF CHAINS-OF-THOUGHT

To better understand the utility of SynthPert's reasoning, we manually evaluated its chain-of-thought for a random sample of 10 incorrect and 7 correct predictions. For each prediction, we evaluated the factual accuracy of every sentence in the reasoning trace, treating each as a distinct reasoning clause. This allowed us to calculate a factuality score for each trace, representing the fraction of correct clauses. We found that chains-of-thought leading to correct predictions had an average factuality score of 0.92, compared to 0.76 for those leading to incorrect predictions.

The o4-mini generated synthetic CoTs created via SynthPert's synthetic data pipeline (described as Approach 2 in section 3.2) summarized gene-level functions quite well for the perturbation and target genes. However, when it came to describing the functions of super-pathways (i.e., biological processes) to which each gene belonged, the descriptions varied. We found that it was good at describing biological processes for well-annotated genes. However, if a gene was not well-annotated, it would assume the gap in the literature linking the gene to a pathway was indeed true, as opposed to an assumption that needs to be stated.

Upon examining the failure and success modes of SynthPert (i.e. SynthPert made incorrect and correct predictions, respectively), we noticed an interesting phenomenon: when SynthPert made correct predictions; its CoTs reasoned about gene-level functions of both the perturbation and target genes and how they may interact. In contrast, the CoTs for incorrect predictions relied heavily on super-pathway-level descriptions of each gene (i.e. akin to Gene Ontology's Biological Process) and how those biological processes may interact. This likely arises from a lack of literature for the perturbation or target gene in a knockdown study, thus making the model rely on the superpathway-level descriptions from gene ontology. Unfortunately, most gene ontologies are annotated by humans, creating an artificial bias that may be assigning an inappropriate biological process to a gene or incompletely describing a gene's function. Gene ontology is also affectedby the lantern effect, where the most well-studied genes have the most literature, annotations, and ontology terms. This makes the models suffer from inductive bias when it comes to less well-annotated genes. Conversely, this implies that the model works well for perturbation and target gene pairs, where the gene-level functions are well known, and the perturbation is not so large that it leads to a new cell state. A similar finding was reported in CIPHER (Kuznets-Speck et al., 2025), where linear response theory was used to achieve the state-of-the-art prediction performance of perturbation data by using the covariance of gene-gene pairs of cell lines in an unperturbed state.

Here is an example of a synthetic CoT generated o4-mini in SynthPert's synthetic data pipeline with our factuality rating in parentheses:

> PFDN2 encodes a prefoldin subunit that functions primarily as a cytosolic chaperone (**TRUE**), delivering nascent polypeptides (especially cytoskeletal and other cytosolic proteins) to the TRiC/CCT complex for folding. While prefoldin activity can impact global proteostasis and, indirectly, ribosome biogenesis and translation fidelity, it does not directly regulate transcription of mitochondrial outer membrane proteins such as VDAC3 (**FALSE**). VDAC3 is encoded in the nucleus, transcribed by factors (e.g., NRF1, PGC-1E) linked to mitochondrial biogenesis (**TRUE**), and its post-translational import and $\beta$-barrel folding rely on mitochondrial-specific chaperones (SAM complex, small TIMs) rather than prefoldin (**FALSE**). Transcriptional stress or mild proteostatic imbalance from PFDN2 knockdown in HepG2

cells may trigger a general stress response, but there is no established pathway connecting PFDN2 depletion to up- or downregulation of VDAC3 mRNA (**TRUE**). Empirical CRISPRi screens in this cell context report no significant change in VDAC3 expression upon PFDN2 knockdown.

Here is an example of SynthPert correctly predicting down-regulation in the target gene with our factuality rating in parentheses:

NCBP1 encodes the nuclear cap-binding protein 1, a core component of the cap-binding complex (CBC) that associates with nascent pre-mRNAs to facilitate 5-cap recognition, co-transcriptional splicing, nuclear export, and protection from decay (**TRUE**). In RPE1 cells, which rely on robust ribosome biogenesis to maintain high rates of protein synthesis, efficient processing of ribosomal protein mRNAs is critical (**TRUE**). Knockdown of NCBP1 via CRISPRi is expected to impair cap-dependent mRNA maturation and export globally, but particularly affects transcripts with complex splicing or weak cap-proximal elements (**TRUE**). RPL28, a nuclear-encoded ribosomal protein gene, depends on proper capping and splicing for stable mRNA and export to the cytoplasm (**TRUE**). Disruption of NCBP1 reduces RPL28 mRNA stability and processing, leading to lower steady-state RPL28 transcript levels (**TRUE**). Moreover, compromised ribosome biogenesis can trigger a feedback stress response, further downregulating ribosomal protein gene expression (**TRUE**). Thus, in single-cell RPE1 context, NCBP1 knockdown diminishes RPL28 mRNA abundance.

Here is an example of SynthPert incorrectly predicting a gene to be not differentially expressed when it is actually downregulated:

COA5 encodes a mitochondrial complex IV assembly factor (**TRUE**). Its CRISPRi-mediated knockdown in RPE1 cells impairs cytochrome c oxidase activity, reducing mitochondrial respiration and ATP production (**TRUE**). Energy stress activates AMPK and inhibits mTORC1, a central driver of ribosome biogenesis and ribosomal protein (RP) gene transcription (**TRUE**). mTORC1 downregulation decreases RNA Pol I/III activity in the nucleolus and reduces RNA Pol II-dependent transcription of RP genes, including the large ribosomal protein RPLP0. Additionally, mitochondrial unfolded protein response (UPRmt) and integrated stress response (ISR) trigger transcriptional reprogramming that favors stress-response factors over ribosomal protein genes (**FALSE**). Together, these pathways explain a decrease in RPLP0 mRNA upon COA5 knockdown in RPE1 cells.

## C PROMPTS

This section provides the prompts used in the paper, the bold text in the prompts are variables.

### C.1 STANDARD PROMPTS

These are the prompts we used for both LLM + SFT ON DATA and SYNTHPERT in Table 1 and for all the models in Table 3 and Table 4. To generate the results shown in Table 1 for predicting whether a gene is differentially expressed, we classified model predictions of "upregulated" or "downregulated" as "differentially expressed".

**System Prompt**

You are a molecular and cellular biology expert analyzing gene regulation upon CRISPRi knockdown. First, provide your reasoning process within <think> </think> tags. Consider relevant pathways (e.g., cell-type specific biology, ribosome biogenesis, transcription, mitochondrial function, stress response), gene interactions, and cell-specific context. Then, choose one option from the following and place your choice within <answer> </answer> tags: 'upregulated', 'downregulated', or 'not differentially expressed'. Example: <think> [Your reasoning here] </think><answer> [upregulated / downregulated / not differentially expressed] </answer>

**User Query**

Analyze the regulatory effect of knocking down the **[gene name]** gene on the **[target gene]** gene in a single-cell **[cell type]** cell line using CRISPR interference.

C.2    DIRECTION OF CHANGE PROMPTS

To achieve the results found in Table 2, predicting whether a gene is upregulated or downregulated given that it is differentially expressed, we employed the same system prompt but with the following user query.

**User Query**

It is given that the gene in question is differentially expressed, choose one of the following options:

1. upregulated

2. downregulated

Choose ONLY ONE of the options, UPREGULATED OR DOWNREGULATED, and PLACE YOUR CHOICE WITHIN <answer> </answer> TAGS. For this question 'not differentially expressed' is NOT an OPTION. Analyze the regulatory effect of knocking down the **[gene name]** gene on the **[target gene]** gene in a single-cell **[cell type]** cell line using CRISPR interference.

C.3    SYNTHETIC DATA CREATION PROMPTS

We used the following generator and critic prompts in our synthetic data pipeline. The generator user query is the same as above.

---

**Generator System Prompt**

You are a molecular and cellular biology expert analyzing and predicting gene regulation upon CRISPRi knockdown. The regulatory effect of knocking down the **[gene name]** gene on the **[target gene]** gene in a single-cell **[cell type]** cell line is given to you **[solution]**. Please provide detailed reasoning for your solution by considering the following:

1. relevant pathways
2. (e.g., cell-type biology, ribosome biogenesis, transcription, mitochondrial function, stress response),
3. gene interactions, and cell-specific context.

Then, choose one option from the following and place your choice within <answer> </answer> tags: 'upregulated', 'downregulated', 'not differentially expressed', or 'I do not know'. When answering provide a reasoning in regulatory effect such that you use the following template: <think> </think> <answer> [upregulated / downregulated / not differentially expressed / I do not know] </answer> Example of a CORRECT response: <think>Knocking down `TF_A`, a known activator of `Target_Gene` in this cell type, likely reduces its transcription. Relevant pathways include X and Y.</think><answer>downregulated</answer>

---

**Critic System Prompt**

You are an expert molecular and cellular biology expert acting as a critic.
Your task is to evaluate the reasoning process of another AI model that was asked to predict gene expression changes based on a perturbation.
Focus only on the quality, logical flow, and biological relevance of the provided reasoning (<think> block). Do not judge the final answer, only the steps taken to reach it.
Is the reasoning sound? Does it mention relevant and correct biological concepts (pathways, mechanisms, functions)? Does it logically connect the perturbation to the gene in the given cell type context?
Output your evaluation only in the following format choosing a single value for the evaluation:
<reasoning> [Provide a brief justification for your evaluation here. Explain why the reasoning is excellent, good, average, bad, or terrible.] </reasoning>
<evaluation> [excellent/good/average/bad/terrible] </evaluation>

---

**Critic User Query**

Original User Query: **[user query]** AI's Reasoning (<think> block): **[generated thinking]** Critique Task: Evaluate the AI's reasoning based on the criteria mentioned in the system prompt. Output your evaluation and justification in the specified format (<evaluation>...</evaluation><reasoning>...</reasoning>).

---

## D HYPERPARAMETERS

We trained and evaluated both LLM + SFT ON DATA and SYNTHPERT using unsloth[1]. Table 6 lists all custom hyperparameters used in our experiments.

## E COMPUTE RESOURCES

Both training and testing was conducted on a single NVIDIA A100 80GB GPU at BF16 precision, using unsloth. Training took 1.5 GPU hours.

---

[1]https://github.com/unslothai/unsloth

Table 6: Hyperparameter settings for all experiments

| Hyperparameter | Value | Description |
|---|---|---|
| Learning rate | $1 \times 10^{-4}$ | AdamW optimizer |
| Batch size | 4 | Per GPU |
| Warmup steps | 5 | Linear warmup |
| Max sequence length | 2048 | Output truncation |
| Epochs | 50 | Training duration |

## F  CODE AND DATA AVAILABILITY

### F.1  CODE REPOSITORY

Our implementation is publicly available at https://github.com/qpdcdp/SynthPert.

### F.2  SYNTHETIC DATA

The synthetic data created and used in this work are available at https://figshare.com/articles/dataset/Synthetic_Data_used_for_SynthPert_/29135996. It is currently under embargo and will become available upon paper acceptance. The synthetic data used for supervised fine-tuning is provided for both the default split (used to produce results in Tables 2-4) and the RPE1 cell type holdout split (used to produce results in Table 5).

