# OpenReview forum: "SynthPert: Enhancing Biological Reasoning in LLMs via Synthetic Reasoning Traces for Cellular Perturbation Prediction"
_ICLR.cc/2026/Conference — ICLR 2026 Conference Withdrawn Submission_

### Official Review · Reviewer_jUeR · 2025-10-25

**Soundness:** 2
**Presentation:** 2
**Contribution:** 2
**Rating:** 2
**Confidence:** 4

**Summary:**

This paper introduces SynthPert, a method for improving LLM performance on cellular perturbation prediction by fine-tuning a base model on synthetic chain-of-thought reasoning traces from a more advanced frontier model. Evaluated on the PerturbQA benchmark, the method reportedly outperforms the frontier model that generated its training data.

**Strengths:**

- The core idea of "synthetic reasoning distillation" is innovative.
- The result that the student model surpasses its teacher is striking if true.

**Weaknesses:**

- The performance of large language models is not comprehensively evaluated in this study. For example, GPT-5-thinking may improve over all proposed models.
- There is a potential data leakage issue. Modern language models evolve in time and are likely to have seen PerturbQA data a priori. Therefore, the risk of data leakage is high, and the performance gain may not be meaningful.
- The novelty of direct 3-way classification is limited.
- The authors used a public GitHub repository to share the code, which is not recommended and can leak author identity. Furthermore, I share the concerns raised in the GitHub issue regarding the metric calculation and method comparison. A satisfactory response is needed to address the reproducibility concern. This is especially important as the authors directly adopt perturbQA results for other benchmarking methods.
- It is unclear how the results compare to some other well-established benchmarks, e.g. [1-2].

[1] Viñas Torné, Ramon, et al. "Systema: a framework for evaluating genetic perturbation response prediction beyond systematic variation." Nature Biotechnology (2025): 1-10.
[2] Ahlmann-Eltze, Constantin, Wolfgang Huber, and Simon Anders. "Deep-learning-based gene perturbation effect prediction does not yet outperform simple linear baselines." Nature Methods (2025): 1-5.

**Questions:**

Please see weaknesses.

---

### Official Review · Reviewer_JZRM · 2025-10-27

**Soundness:** 3
**Presentation:** 3
**Contribution:** 3
**Rating:** 4
**Confidence:** 5

**Summary:**

This paper tackles the problem of predicting perturbation effects using LLMs. The input is a textual description of the cell type, perturbation, and target gene, and the label is the direction of change in expression for the input gene. In addition to the no-context approach, the authors introduce a new fine-tuning strategy that involves training the model using synthetic, biological explanations of observed train labels. They demonstrate that the latter approach outperforms existing methods across four cell lines.

**Strengths:**

- The paper introduces an effective knowledge distillation scheme via SynthPert and shows that training on reasoning traces outperforms training on data alone. This has implications for future tuning of LLMs.
- The paper rigorously evaluates the model's ability to generalize across biological contexts by performing an experiment where an entire cell line  (RPE1) was held out from training. RPE1 is the only non-cancerous line among the four, and the results suggest the model can transfer biological principles from cancer cell lines to healthy.

**Weaknesses:**

- The authors reference works that show that simpler baselines often outperform complex models, but such baselines are missing from the current study. The appropriate train/cell-specific mean/majority baselines (e.g., as done in STATE [1]) should be included in all experiments to further verify that the results are not due to memorization.
- The authors do not specify what downstream ML model was trained on the GenePT/scGPT embeddings to produce results in tables 1-2. Did the authors experiment with multiple model families and were the hyperparameters tuned? Addressing the choice of the underlying predictor is necessary to assess the rigor of the baseline comparisons.

[1] https://www.biorxiv.org/content/10.1101/2025.06.26.661135v2

Other
- In your last example in Appendix B, it seems from the model output that the gene is being downregulated, but you claim the model incorrectly predicts this gene to not be DE.

**Questions:**

The process of generating a large body of synthetic data and then filtering it down by 98% using a judge LLM implies a significant computational cost. Can the authors comment on the resources required for this step and accessibility to the general public?

---

### Official Review · Reviewer_wrsK · 2025-10-30

**Soundness:** 3
**Presentation:** 3
**Contribution:** 2
**Rating:** 4
**Confidence:** 4

**Summary:**

This paper introduces SynthPert, a method for predicting cellular responses to genetic perturbations using large language models (LLMs). The task is: given (cell type, perturbation, target gene), predict whether that gene will be upregulated, downregulated, or not differentially expressed after the perturbation.
Key ideas:
1. Instead of just fine-tuning a model on labeled tuples, SynthPert fine-tunes on synthetic chain-of-thought (CoT) biological explanations generated by a stronger "frontier" LLM (o4-mini).
2. These explanations are filtered by a critic LLM and only “excellent”-quality reasoning traces are kept.
3. A smaller open model (DeepSeek-R1 8B, LoRA-tuned) is fine-tuned on those selected reasoning traces.

They evaluate on PerturbQA, which reformulates perturbation biology as a natural-language classification task. Compared to baselines including GEARS, scGPT, GenePT, SUMMER (RAG + LLM reasoning), and a plain SFT-on-data baseline, SynthPert:
1. Achieves higher AUROC on differential expression and direction-of-change across 4 human cell lines.
2. Solves the full 3-class prediction problem directly (up / down / no change), rather than splitting into two subtasks like SUMMER.
3. Generalizes to an unseen cell type (RPE1) with ~87% accuracy, suggesting cross-cell-type transfer.
4. Beats its own teacher model (o4-mini) on that 3-class task (89% vs 52% accuracy in the general setting), despite being smaller and trained on only ~2% of filtered data.

The authors argue that reasoning structure itself, even if partially factually wrong, is more valuable than raw labels, and that synthetic reasoning distillation can “unlock” latent biological capability in mid-size LLMs.

**Strengths:**

1. Strong empirical results
SynthPert consistently outperforms strong baselines (SUMMER, GEARS, scGPT, GenePT, GAT) under PerturbQA.
It works in a challenging 3-class joint setup, not just binary subtasks.
It generalizes to an unseen cell line — which is a big deal, since cross-cell transfer is usually where perturbation models break.

2. Surpassing its own teacher
This is eye-catching: the student (8B, LoRA-tuned on filtered traces) beats the teacher frontier model on the final task. That’s the kind of result ICLR likes because it suggests an interpretable path toward smaller, specialized domain LLMs without continuously calling an expensive frontier model.

**Weaknesses:**

1. Reliance on proprietary frontier models
The synthetic traces come from o4-mini (frontier model) and are filtered by another LLM “critic.” This raises some questions:
How reproducible is this pipeline for other labs that don’t have access to those same frontier models?
Would an open model judge produce equally good traces?
Is there a risk of just laundering proprietary model knowledge into a smaller open(ish) model?

2. Biological correctness vs “sounding biological”
The critic LLM is grading explanations on being “biologically plausible,” but plausibility ≠ correctness. The appendix admits that even high-scoring chains-of-thought still include false mechanistic statements, especially at the pathway level, and that “hallucinated pathway links” show up more in wrong predictions. That means the model can still be confidently wrong — in a way that sounds mechanistic.

3. Distributional skew / class imbalance
Most genes are “not differentially expressed,” so a naive model could try to game accuracy by spamming that class. The authors argue SynthPert is not just exploiting that, and they do mention class rebalancing experiments. Still:
Upregulated recall for SynthPert (0.14) is low compared to o4-mini (0.62), even though SynthPert has higher precision.
For discovery work, low recall on minority classes can be a practical issue: you miss interesting positive hits.

**Questions:**

1. Distillation Paradox
The smaller student model surpasses its larger teacher.
→ What exactly allows this “distillation paradox”?
→ Does it arise from regularization (LoRA compression), or from filtering low-quality traces that reduce noise?

2. Synthetic Data Generation
The critic model uses a 5-point scale to rate explanations.
→ What fraction of generated reasoning traces were rejected vs. retained?
→ How sensitive are final results to the critic’s strictness (e.g., “excellent” vs. “good” threshold)?
Were hallucinations or factual errors in the remaining “excellent” traces quantified beyond the small 17-example sample in Appendix B?

3. Cross-Cell-Type Generalization
The RPE1 holdout experiment is impressive (87% accuracy), but RPE1 is epithelial and differs substantially from training lines.
→ Has generalization been tested on other withheld lineages (e.g., immune or neuronal cells)?
→ Could this success stem from overlapping perturbation targets rather than true cross-lineage generalization?

---

### Official Review · Reviewer_xpVd · 2025-11-04

**Soundness:** 2
**Presentation:** 2
**Contribution:** 2
**Rating:** 2
**Confidence:** 5

**Summary:**

This paper introduces SynthPert, an LLM based approach for predicting cellular perturbation effects. The key idea in this paper is to fine-tune an LLM on synthetic chain-of-thought reasoning generated by OpenAI o4-mini. They show that SynthPert achieves state-of-the-art  results on the PerturbQA benchmark.

**Strengths:**

- SynthPert achieves SOTA results on PerturbQA
- SynthPert shows a practical way to use LLMs for predicting cellular perturbation effects

**Weaknesses:**

- The overall contribution of this paper is incremental. The proposed method, fine-tuning a language model on synthetic reasoning traces generated by a larger model, follows a standard pattern of LLM distillation and reasoning supervision, without introducing a new architectural insight, training paradigm, or theoretical framework. The approach is interesting from a practical perspective, but conceptually it represents a straightforward application of existing ideas (e.g., chain-of-thought supervision). This paper is more suitable for a workshop paper rather than a full conference.
-The empirical scope is limited. The evaluation relies solely on the PerturbQA benchmark, which is limited.
- PerturbQA is a derived benchmark published in 2024 (Wu et al.). Given o4-mini’s pretraining cutoff is likely late 2024 or early 2025, it’s possible that the benchmark’s data distributions, example tuples, or even results tables were present in its pretraining corpus.

**Questions:**

Please see above.

---

### Note · Authors · 2025-12-03

**Comment:**

⁠We sincerely thank the reviewers for their thoughtful feedback and the time invested in evaluating our work. After careful consideration, we have decided to withdraw our submission while we revise the manuscript for future submission.

**Withdrawal Confirmation:**

I have read and agree with the venue's withdrawal policy on behalf of myself and my co-authors.